# An Exploratory Study of the Critical Success Factors of the Global Shipping Industry in the Digital Era

Surinder Brrar [1,*], Eunha Lee [2] and Tsz Leung Yip [1]

1 Department of Logistics and Maritime Studies, The Hong Kong Polytechnic University, Hong Kong, China
2 College of Natural Sciences, Korea National Open University, Seoul 03087, Republic of Korea
* Correspondence: ssbrrar@polyu.edu.hk

**Abstract:** The global shipping industry faces many uncertainties which impact on how organisations within this sector will perform in the future. Research in the critical success factors which impact the global shipping industry in the digital era is lacking. This study plugs the gap in the literature by identifying four key critical success factors which are innovation capability, risk governance capability, leadership and strategic capability, and technological capability. In addition, this study also found three organisational performance measures that are useful for senior management teams within the industry, namely, financial performance, operational performance and marketing performance. The results were then triangulated and validated by the case study method using a global shipping organisation. The findings establish a set of critical success factors and the corresponding relationships between the identified critical success factors and the identified organisational performance measures. The paper also provides managerial insights for industry practitioners for defining, prioritising and allocating resources in order to improve organisational performance.

**Keywords:** critical success factors; organisational performance; shipping industry; digitalisation; digitisation





## 1. Introduction

Over the past 50 years, global maritime trade volumes have grown remarkably with trade volumes moved by shipping having the largest share of global international trade, with its share ranging between 80 and 90 percent [1]. Shipping is the most practicable and lowest cost way of transporting large volumes of cargoes in global trade [2]. The global shipping industry is a complex and uncertain business with the crucial role of facilitating international trade and global economic development. However complex and uncertain it may be, all different segments of the global shipping industry face a multitude of further complications which will impact on how the sector performs in the future. Emerging issues and opportunities are challenging norms and may even be the catalysts that could set new paths for the sector [3].

The introduction of new technology in the global shipping industry will attract unanticipated and new market entrants, creating new and higher expectations in customers, which in turn open opportunities for new business models to evolve. Digitalisation is driving the maritime industry beyond its traditional ways of doing business and can provide new opportunities to enhance productivity, improve efficiency and contribute positively to the sustainability of the logistical systems [4–7].

Much of the prior research has been made on relationships which are limited to an operational focus, a narrow and specific focus area such as market development, or based on other subsectors of the shipping industry. While digital transformation is a topical theme in shipping research and professional practice [6] and there is an increasing interest in this subject, arising from the necessity of raising the global shipping industry to the same level of digitalisation as other industries [8], there is thus a gap in the literature on research which this paper addresses. The research questions in this study are:

1. What are the critical success factors of organisations of the global shipping industry in the digital era?
2. What are the organisational performance measures of the global shipping industry in the digital era?
3. What are the relationships between the critical success factors and organisational performance measures identified in the study?

## 2. Literature Review and Theoretical Background

Bullen & Rockart [9] developed a valuable study on 'A Primer on Critical Success Factors' which builds on an earlier study introduced in the Harvard Business Review article titled 'Chief Executives Define Their Own Data Needs' [10]. Rockart [10] claimed that "critical success factors are, for any business, the limited number of areas in which results, if they are satisfactory, will ensure successful competitive performance for the organization." Rockart [10] also argued that different firms will have different critical success factors and they could be ranked by their importance to the individual organisations. Leidecker & Bruno [11] posited that critical success factors are "important and indispensable", and they can foster organisations to achieve their goals in operation. Using critical success factors as a set of key criteria for analysing the strengths and weaknesses appraisal of an organisation (including competitor organisations) also plays a key role in the long-term planning and strategy development process of an organisation. Thus, critical success factors are crucial variables for organisations to achieve their objective or success of the business and further to maintain their competitive advantages.

In shortlisting the critical theories that are the key theoretical support for this paper we concluded that six theories closely connected to the study of the critical success factors and their relationship with organisational performance measures are particularly relevant. These are (1) Porter's generic strategies; (2) resource-based view (RBV); (3) competence-based theory; (4) dynamic capability theory; (5) resource dependence theory and (6) organisational culture theory.

Porter [12,13] put forward his theory that in order to achieve strong organisational performance organisations must employ one of these three generic strategies: (1) cost leadership; (2) differentiation focus; or (3) strategic focus. The adoption of these three generic strategies was linked to organisational performance in later studies [14,15]. Darrow, King and Helleloid [16], for example, identified successful strategic options for smaller hardware enterprises to remain successful in face of an onslaught of larger hardware organisations, and concluded that by using critical success factors in executing their business plans many small and independent hardware stores can continue to remain successful in the changing market dynamic.

The resource-based view has developed as a managerial framework that establishes that capabilities and resources are relevant and important for better understanding of the sources of sustainable competitive advantage for an organisation [17]. The resource-based theoretical model of sustained competitive advantage includes those organisational resources which are important and relevant, uncommon, not imitable in a perfect way and cannot be substituted easily [18]. Yang and Lirn [19] analysed the resource-based view on the logistics and shipping industry performance and found that the resources and relationships of an organisation support are valuable in enhancing shipping organisational performance in terms of superior logistics service capabilities. In a shipping company context, Yuen et al. [20] demonstrated that the resource-based view model is validated in terms of finding that the effects of sustainable shipping capabilities on business performance are dependent on the environment of a shipping organisation, both externally and internally.

Competence-based theory is a systemic and structured method of thinking on how an organisation can achieve high performance over an extended period of time. It integrates the economic capability, resource availability, and behavioural systems and values of organisations in a theoretical framework which is dynamic, systemic and holistic [21]. Freiling [22] defined competence as something within the organisation which is repeatable

and learning-based. Further, Freiling [22] posited that this theory connects the organisation to the market and thus differentiates itself from the resource-based view that the sustained competitive advantage comes from exploiting the resources of the organisation.

Dynamic capability theory is not only required for capability building and strategic change, but it also addresses the big question of how sustaining a capabilities-based advantage in the changing dynamics of an organisation's environmental situation can be successfully implemented [23]. Teece [24] concluded that fast-paced business environments, that are open to global competition and characterised by dispersed organisational and geographical sources of demand, supply and innovation, require distinct and difficult-to-copy dynamic capabilities that can be used for the enterprise's unique asset base to build, explore, improve, protect, and keep relevant and fit for strategic use in the environment the organisation is in. Recent shipping industry research undertaken [25–27] has identified the existence of strong connections between organisational, process innovation and dynamic capabilities in the shipping sector.

Resource dependence theory places the organisation in an open system with its relating dependence on the external environment [28,29]. It recognises that external factors have an influence on organisational behaviour and that management can act to reduce such dependence by reducing other's power over them [30]. In terms of the global shipping industry, it points to the notion that an organisation's need for resources provides opportunities for others to gain control over it [31,32]. This theory shows that organisations will have more power relative to others to the extent that they have control of the resources needed by others and whether they are able to mitigate their reliance on others for their needed resources [33].

Denison & Mishra [34] found that the culture of organisations has an important influence on its effectiveness. Schein [35] stated that culture is the most difficult organisational factor to change. It outlasts almost everything in the organisation, including its founders, its products and services, its leadership, and its physical attributes. Schein [27] further attributed the organisational model from the viewpoint of an observer, describing organisational culture from three standpoints. Firstly, artefacts, which are the critical components of an organisation containing cultural meaning. In a shipping context, these artefacts would comprise awards and prizes won, large models of the organisation's ships placed prominently where staff and customers can see. Secondly, rituals, where the collective organisational behaviour constitutes the fabric of the organisational culture (the ringing of the 'fixing' bell on the special occasions upon fixing a shipping contract or achieving an outstandingly favourable voyage fixture). Finally, at a deeper level, an organisation's culture is where an organisation's tacit assumptions are found. If sufficiently enforced by the top management of an organisation, the culture of an organisation can be a powerful influence on the pace of innovation in the digital era.

There has been limited amount of research done on the critical success factors required in the shipping industry and there has not been any research carried out thus far on the critical success factors of the global shipping industry in the digital era, which is the focus of this study. An examination of critical success factors in relation to quality management can provide very good insights for other fields such as the global shipping industry in the digital era [36]. Badri, Davis and Davis [37] and Saraph, Benson and Schroeder [38] developed instruments that assist in identifying the critical factors required to measure organisational improvement in the context of quality management. Shipping is an atomistic, highly fragmented market which is complicated further by the rise of digital technologies, which has led to several key innovations within the industry, including blockchain-driven innovations, streamlined cargo documentation flows and integrative logistics [39]. Ichimura, Dalaklis, Kitada and Christodoulou [40] concluded that major shipping companies have embraced digitalisation to increase cost efficiency, raise competitiveness and meet the needs of their customers. Earlier, Nikitakos and Lambrou [41] considered emergent digital shipping modes of operation and important determinants of an organisational decisional context as essential means in order to set digital shipping strategies, design market policies, and

design and implement business models and technical options towards a future frictionless and networked shipping environment.

There is a wide extant body of literature that details the importance of having a critical success factor method to identify the key areas in which managers must have favourable results in order to achieve their goals [9–11,42–44].

Like most industries in the digital era the global shipping industry is currently facing unprecedented level of changes. Reiedl et al. [45] described the anticipated digital disruption in the following form: start-ups developing digital business models focused completely around the customer; incumbent competitors rapidly digitising their business models; suppliers, such as shipping lines, rapidly digitising their booking process; integrators rapidly expanding their reach by leveraging their end-to-end information technology systems; and customers with a strong technological supply chain capability.

Organisational performance measures are used in achieving business goals and objectives by the employment of appropriate critical success factors. This requires a set of criteria to measure both in a qualitative and quantitative manner an organisation's financial and non-financial performance. Return on investment (ROI) or return on equity (ROE), cost of overheads per vessel-day, time-charter-equivalent (TCE or the earnings on the vessel per day) and performance against the relevant market indices are important organisational performance measures within global shipping companies. Otheitis and Kunc [46] found that early adopters of a performance measuring system are among the industry leaders in shipping. Sanchez-Gonzales, Diaz-Gutierrez and Nunez-Rivas [47] suggested using the four performance dimensions of costs, time, quality and flexibility to identify key performance indicators applicable for tracking performance.

As explored in the extant literature and consultancy reports, the global shipping industry is facing rapid, innovative and potentially disruptive development of digital tools and measures in improving commercial and operational efficiencies facilitated by digitalisation. It could result in a Schumpeterian creative destruction of the global shipping industry business model as we know it today and benefit the industry by combining its resources in an integrative sense, of product, process, organisational, market innovation and new sources of data [48].

## 3. Research Framework and Hypothesis Development

Initial hypotheses to identify the critical success factors of the global shipping industry in the digital era are examined between the relationships between the initial constructs of the critical success factors and the organisational performance measures of the study. The framework has been designed to address the three research questions, namely (i) what are the critical success factors of organisations of the global shipping industry in the digital era, (ii) what are the organisational performance measures of the global shipping industry in the digital era, and (iii) what are the relationships between the critical success factors and organisational performance measures identified. The initial constructs and measurement indicators were identified by reviewing the extant and relevant literature available, in the (i) generic and (ii) industry-specific critical success factors and organisational performance measures. These are then utilised in the formulation of hypotheses in the study as shown in Figure 1.

A large number of consultancy reports were also needed to be reviewed and included so that the most up-to-date practical information for the constructs and their corresponding measurement indicators could be developed. Through a quantitative approach both critical success factors and their relationships with organisational performance measures pertinent to the global shipping in the digital era were identified. Data collection was carried out via a structured survey questionnaire with appropriate statistical tests of data reliability and validity utilised to rigorously find and test the results thus obtained. Later, exploratory factor analysis (EFA) was used to build a list of constructs of potential critical success factors and organisational performance measures that are relevant to the global shipping industry in the digital era in this study. Further, multiple regression analysis (MRA) was

utilised in finding out the relationships that exist between the identified capabilities and the identified organisational performance measures relevant to the global shipping industry in the digital era.

| **Critical Success Factors** | **Quantitative Study** | **Organisational Performance Measures** |
|---|---|---|
| • Critical Success Factors of the Global Shipping Industry in the Digital Era<br><br>• Multi-faceted perspective from both extensive literature review and consultancy reports | • Structured survey questionnaire<br><br>• Pilot experiment of questionnaire<br><br>• Expert validation of questionnaire<br><br>• Targeted respondents: senior executives of Shipping companies | • Multi-faceted perspective from both extensive literature review and consultancy reports<br><br>• Organisational Performance Measures Pertinent to Global Shipping Industry in the Digital Era |

**Figure 1.** General research framework.

Detailed relevant and extant studies have been conducted in order to prepare a set of initial constructs of the critical success factors of the global shipping industry in the digital era. Only the initial constructs of critical success factors that were common, core and pertinent to this study were considered. As this is an area with relatively limited prior research, the initial constructs were defined by identifying and then combining the following research categories: first, the critical success factors of the service industry with emphasis on the service and transportation industry were identified; second, the critical success factors of the global shipping industry; and third, the critical success factors in the digital era were then identified. Using a process to benchmark the generic constructs of the critical success factors of the transportation and shipping industry can assist in broadening the scope in order to develop and identify the constructs of the critical success factors in the global shipping industry in the digital era.

Upon the identification of the eight initial constructs of the critical success factors, the corresponding measurement indicators of each initial construct of the critical success factors were identified using these principles: the most relevant and appropriate measurement indicators were selected according to what is 'most common, core and critical'; measurement indicators were directly adopted if they directly suit the objectives of the study; keyword or paraphrased measurement indicators were combined for those with similar meanings; appropriate renaming of measurement indicators was performed to fulfil the purpose of the initial constructs of the critical success factors; and new measurement indicators were added according to the factual real-world business needs and environment. To check for suitability and validate their appropriateness for this study, all measurement indicators went through a content-validation process whereby three academicians and two industrial practitioners examined the indicators thoroughly.

In terms of researching business policy, organisational performance is an important component of empirical research and researchers often take the performance of organisations into consideration when examining organisational strategy, structure and planning processes [49]. Venkatraman and Ramanujam [50] posited that performance improvements are the base of strategic management since performance is a time-tested result of any strategy. Organisational performance is the most appropriate dependent variable for most areas of management. Customer dynamics, supply resources, capital availability and employees make organisational performance fundamental for modern organisations to obtain success and durability [51]. However, getting a framework of performance measurement measures

that can be relevant, cost-effective and replicable on a regular basis with respect to a specific organisation is rather more challenging. Moreover, there is always a risk that the measures chosen could lead to undesirable negative consequences as well as possible higher order effect or effects that were completely unexpected.

This study tested eight hypotheses in line with the objectives of examining possible relationships between the constructs of exogenous origins (i.e., critical success factors) and the constructs of endogenous origins (organisational performance measures).

Leadership has considerable influence on the strategic capabilities of an organisation due to the fact that the leadership team is tasked with the role of determining the vision, mission and objectives and decides on a strategic plan of action in order to achieve it. Prahalad [52] argued that top managers should focus their attention and time on developing strategic capability, which is defined as the ability for an organisation to think, act and implement desired actions that are strategic in an ever-changing competition-driven external environment instead of using the resources of an organisation in pursuing random and fashionable fads. In a shipping context, Jenssen [53] argued for an increased focus for organisations in improving their strategic capability in order to survive in a high-cost environment by creating distinctive competitive advantages that are difficult to copy. As a consequence, leadership and strategic capability are likely to be among the critical success factors of the global shipping industry in the digital era. The first hypothesis, H1, is constructed as follows:

**H₁.** *Leadership and Strategic Capability is Positively Associated with Organisational Performance of the Global Shipping Industry in the Digital Era.*

The highly competitive global shipping industry requires the specific shipping organisation to provide high-quality service levels to ensure customer satisfaction, opportunity for repeat business, preference in long-term contract opportunities and opportunities for growth in new business generated by customers. Quality of service has a strong association with customer satisfaction. Yuen and Thai [54] found that customer satisfaction and quality of service (such as speed, reliability, responsiveness and value) are strongly correlated in a related shipping study. Hirata [55] found that the top three service characteristics influencing customer satisfaction, in a related container liner shipping context, are as follows: first, the quality of the customer service representative; second, the quality of digitalisation initiatives carried out; and third, the quality of the sales representative, in that order. Hirata [55] further found that customers expect every organisation to deliver products and services with a seamless user experience, demanding digital business processes with intuitive user interfaces, around-the-clock availability, tailored and personalised treatment, globally consistent throughout the organisation and with zero errors. Digitalisation topped the customer's agenda in measuring customer satisfaction. Therefore, Hypothesis 2 (H2) is formulated as follows:

**H₂.** *Service Quality is Positively Associated with Organisational Performance of the Global Shipping Industry in the Digital Era.*

As a driver of innovation and a key organisational resource, technological capability can bring about product and process improvement and re-engineering, an increase in efficiency, a reduction in costs, etc. [18,56]. Organisations which do not foster and grow their technological capabilities will gradually see this erode in the onslaught of competitors growing faster than them. These organisations will often lose market share, which may then either (i) reduce pressure and allow for their recovery or (ii) lose critical resources and face further downward pressure [52]. In the wider transportation scope, technological innovation would likely be in the information technologies where better utilisation of existing assets and resources will derive productivity gains [57] using reporting and dashboard tools such as PowerBI, Tableau and many other similar business performance analysis systems. The shipping industry has been slow to adopt digital technologies but is quickly

catching up with 'smart shipping' (i.e., networks of connected ships) which is regarded as the next phase of the shipping industry's development [58,59]. When used in a systematic framework, digitisation can obtain real-time analytics results that support real-time vessel operations, support better maintenance planning for the vessel and help to facilitate new and better designs in the future, considering lessons learned while designing new components and for the shipbuilding process [60]. Supported by these justifications, technology capabilities are widely recognised as critical success factors in the global shipping industry in the digital era.

**H₃.** *Technology Capability is Positively Associated with Organisational Performance of the Global Shipping Industry in the Digital Era.*

Organisations will need to innovate as their business environment becomes more dynamic and their competitive dynamics evolve when new business models, substitutes for their product or service come to the market, and new entrants threaten to reduce their margins. They also need to innovate as their customers change their demands and lifestyles as well as to capture opportunities offered by new technologies and changing marketplace, business dynamics and market structures [61]. Baregheh et al. [61] further posited a multi-discipline definition of innovation to be the multiple processes where organisations can make new ideas into something which will bring value to them in new and improved services, products and processes. They do this in order to improve their capabilities, compete better for improved pricing power, advance and improve their business model as well as to find ways to differentiate themselves. Camison and Villar-Lopez [62] concluded that the results show that organisational innovation supports the resource-based view that organisational innovation can be seen as valuable, rare, substitutable, durable, not easy replicated and suitable as sources of an organisational competitive advantage. In a shipping context, the shipping industry is slowly going through a shift in industry technology dynamics, with smart ports, smart connected ships and connected logistics infrastructures. Big data analytics technology capability is fast becoming a key part for both the on-board vessel data sources and the shipping company office where the big data analytics capability is often based [63].

**H₄.** *Innovation Capability is Positively Associated with Organisational Performance of the Global Shipping Industry in the Digital Era.*

Organisations with a sustainable financial performance commonly possess a good framework of managerial values core to that organisation that fosters innovativeness and flexibility [64]. Delaney and Huselid [65] explored the links between training and staffing selectivity and found positive associations with perceptual organisational performance measures. Human capital development is important in terms of current technologies, e-learning initiatives, and knowledge management in general, especially in digitisation initiatives such as big data analytics, robotic technologies, 3D printing, drone technologies and Internet of Things. These facilitate a continuous flow of information which could be useful for improving decision making [63]. The resource-based view framework, when examined in relation to human resource management within the shipping industry, considers both the shore-based staff and the sea-based staff as a core resource of the organisation [57]. Empirically, studies have shown [66–68] that human resource management is significantly associated with organisational performance. Organisational flexibility also means that the organisation has the senior management capability to effectively control resources and ensure the organisation's structural flexibility is maintained.

**H₅.** *Organisational Flexibility is Positively Associated with Organisational Performance Measures of the Global Shipping Industry in the Digital Era.*

To achieve competitive advantage, organisations will need to develop strategic capabilities in several key areas. These capabilities must be difficult to copy in order to be sustainable. They should also be in alignment with and be in support of the organisation's strategy [18]. One of the key areas is in an organisation's marketing capability because of the important role of market selection and because of their ability, in a fundamental way, in the implementation of an organisation's market strategies. As such, the marketing capability of an organisation is part and parcel of an organisation's fundamental core capability which supports strategy implementation and hence impacts on organisational performance. Vorhies [69] found that businesses with the highest degree of marketing capability outperformed those with less organisational effectiveness. Morgan, Vorhies and Mason [70] examined the links bounded by market orientation with marketing capabilities and found that both contribute to superior organisational performance. In a shipping context, Panayides [71] found that organisations which only pursue absolute cost advantages may not achieve superior performance, rather, organisations with a dedicated and focused market orientation can be more successful in being high performers.

**H6.** *Marketing Capability is Positively Associated with Organisational Performance of the Global Shipping Industry in the Digital Era.*

The shipping industry is inherently a risky business, with its cyclical nature, long and brutal down cycles, and rather short upcycles. Large fortunes are regularly made within shipping with exceptional speed and, correspondingly, previously large fortunes are being destroyed regularly. Processes for effectively and consistently setting and managing the strategy of a shipping company, with its risk appetite and return targets, and arriving at common ground when several of the decision-makers are involved is complex as each shipping deal has its own risk characteristics [72]. Andreou, Louca and Panayides [73] found in a study on maritime organisations that important corporate governance measures such as the number of members on the board, the presence of robust corporate governance committees, insider ownership, the number of directors as a percentage serving on the boards of other organisations and CEO/Chairman duality, whereby the CEO also acts in a second role as the Chairman of the board, have significant associations with organisational performance. In contrast, Syriopoulos and Tsatsaronis [74] found differing financial performance responses to common governance practices.

**H7.** *Risk Governance Knowledge is Positively Associated with Organisational Performance Measures of the Global Shipping Industry in the Digital Era.*

Rapid digitalisation is prompting the shipping industry to go beyond its old limits and can provide plentiful opportunities to improve efficiency, productivity and the potential sustainability of its performance. Key to this is the facilitation role which is played by digitalisation and its related technological investments, methodological difference in the applications for a better platform of cooperation in information sharing as well as in better coordination and collaboration between counterparts [5]. Using Porter's Five Forces Model [75–79], the composition and strength of the five forces in total determine the nature of the competition within an industry and the average profitability for the incumbents.

**H8.** *Digitalisation Capability is Positively Associated with Organisation Performance Measures of the Global Shipping Industry in the Digital Era.*

## 4. Methodology

The research methodology steps were as follows: identification of the initial constructs and their corresponding measurement indicators of critical success factors and organisational performance measures of this study; development of the structured survey questionnaire; and exploratory factor analysis—setting up of the factor model of new sets of constructs of critical success factors and organisational performance measures followed

by analysing relationships between the potential critical success factors and organisational performance measures pertinent to the global shipping industry in the digital era. Based on the results from the exploratory factor analysis, the eight initial hypotheses were revised to examine and predict the relationships between the potential critical success factors and the respective organisational performance measures. These relationships were then determined using multiple regression analysis (MRA). The relationships were identified by using the following formula:

-   $OPM_i = a_i\ CP_1 + b_i\ CP_2 + c_i\ CP_3 + \ldots \ldots + k_i\ CP_n + C_i + R_i$ where
-   $OPM_i$ shows constructs of organisational performance measure ($i = 1,\ldots, n;$)
-   $CP_i$ shows constructs of critical success factors ($i = 1,\ldots, n$);
-   $a_i, b_i, c_i, \ldots \ldots, k_i$ reflect regression coefficients;
-   $C_i$ represents the intercept/constant value ($i = 1, \ldots, n$); and,
-   $R_i$ represents the error term ($i = 1, \ldots, n$).

The research was carried out at an organisational level due to the nature of the research topic of the study. As this study is of an exploratory study, no prior and extant survey instrument was available. Therefore, cross-sectional data was collected by the usage of the structured survey questionnaire specifically designed for this study [64]) which utilised exploratory factor analysis and multiple regression analysis. The associated relationships between the identified factors and measures are hence appropriately identified by exploratory factor analysis and were then weighed by multiple regression analysis methodology.

## 5. Results

Among the 532 (out of about 1900 BIMCO members who were selected based on accessible senior management contact information from publicly and privately available sources) disseminated structured survey questionnaires sent out by email there were 109 completed responses suitable for analyses. This yielded a response rate of approximately 20.5% which is considered adequate for organisational-level research targeting very busy senior management of the respondent firms [80]. The results indicate that a very high percentage of the respondent organisations viewed critical success factors as essential in their drive towards organisational performance and survival, and as a competitive advantage to outperform business rivals and competitors. The 109 returned and valid responses were divided into two groups according to whether they responded to the first wave of the structured survey questionnaires (*n* = 71, approximately 65.1% of the total returned and valid responses) or the second significant wave structured survey questionnaire (*n* = 38, approximately 34.9% of the total returned and valid responses). A non-response bias test [81] was conducted on the two groups in order to examine the differences in mean values of the critical success factors of the global shipping industry in the digital era. The observation was that there were no significant differences in the mean values regarding the respondent organisations' responses to the constructs of the critical success factors or their corresponding measurement indicators of the global shipping industry in the digital era. This indicates that the non-response bias did not influence the survey data collected from the 109 returned and valid responses in the two significant waves of the survey questionnaire. This study (Appendix A) then conducted two tests, the Kaiser–Meyer–Olkin measure of sampling adequacy (KMO) and Bartlett's test of sphericity (Bartlett's test), to gain confidence that the retained measurement indicators were appropriate for the next step of the exploratory factor analysis (EFA). The KMO value of the retained measurement indicators of the critical success factors of the global shipping industry in the digital era was 0.818, which is higher than the threshold of 0.5 and reached the minimum acceptable level for EFA [82,83].

The organisational performance measures KMO value was 0.863. As such, the retained measurement indicators of organisational performance measures were also appropriate for EFA [82,83]. The corresponding value of Bartlett's test was 1311.543 at a significance level of 0.000. The results suggest that the correlation matrix was not an identity matrix and that the inter-correlation matrix contained sufficient common variance [82]. EFA was conducted

on the 44 measurement indicators. For the extraction method, principal component analysis with varimax rotation with Kaiser normalisation was employed, retaining factors with loadings ≥0.50 and cross-loadings ≥0.40 [84]. From the results of EFA a total of 8 retained measurement indicators of critical success factors were eliminated from the 25 original retained measurement indicators, leaving 17 measurement indicators of critical success factors in a four-factor model for further analysis. EFA eliminated a total of 4 retained measurement indicators from the 19 original retained measurement indicators, leaving 15 measurement indicators of organisational performance in a three-factor model for further analysis. With the factor loadings of ≥0.50 and eigenvalues of ≥1, a four-factor model of critical success factors was derived from EFA.

After the four-factor model was extracted by EFA, the results of the model with the corresponding 17 measurement indicators were construed by assigning labels to them. On account of the nature and semantics of the measurement indicator descriptions under the respective factors, it is theoretically acceptable and meaningful to arrange the measurement indicators under the same construct. The above revised set of labelled constructs, representing the potential critical success factors of the global shipping industry in the digital era, were used as the new constructs of the independent variables in the subsequent multiple regression analysis (MRA) for this study. On the basis of the same criteria (factor loadings ≥0.50 and eigenvalues ≥1), a three-factor model of organisational performance measures with 15 measurement indictors was derived from EFA. After the three-factor model was extracted by EFA, the resulting model with the corresponding 15 measurement indicators were construed by assigning labels to them. On account of the nature and semantics of the measurement indicator descriptions under the respective factors, it is theoretically acceptable to arrange the measurement indicators under the same construct.

The revised set of labelled constructs, representing the potential critical success factors of the global shipping industry in the digital era, were used as the new constructs of the dependent variables in the subsequent multiple regression analysis (MRA) for this study. The CITC (corrected item-total correlation test) and reliability test were employed to determine which measurement indicators could be eliminated in order to purify the data. Out of the 44 original measurement indicators (25 from critical success factors of global shipping industry in the digital era and 19 from organisational performance measures), 32 indicators remained while 12 were eliminated as a result of CITC, reliability test and EFA. Accordingly, a revised set of three new hypotheses were established as discussed earlier using the revised construct labels and are summarised as follows:

**Hypothesis 1 (NH$_1$).** *Innovation capability (INC), risk governance capability (RGC), leadership and strategic capability (LSC) and technological capability (TEC) are significantly associated with financial performance (FIP).*

**Hypothesis 2 (NH$_2$).** *Innovation capability (INC), risk governance capability (RGC), leadership and strategic capability (LSC) and technological capability (TEC) are significantly associated with operational performance (OPP).*

**Hypothesis 3 (NH$_3$).** *Innovation capability (INC), risk governance capability (RGC), leadership and strategic capability (LSC) and technological capability (TEC) are significantly associated with marketing performance (MKP).*

The results of the MRA depict the relationships between critical success factors of global shipping industry, that are innovation capability (INC), risk governance capability (RGC), leadership and strategic capability (LSC), and technological capability (TEC); and organisational performance measures relating to financial performance (FIP), operational performance (OPP) and marketing performance (MKP) of the global shipping industry in the digital era. The results confirmed that all exogenous constructs of the potential critical success factors of the global shipping industry in the digital era are significantly associated

with the endogenous constructs of the organisational performance measures pertinent to the global shipping industry in the digital era and are summarised as follows:

- *Innovation capability, risk governance capability and leadership and strategic capability are significantly associated with financial performance.*
- *Risk governance capability and technological capability are significantly associated with operational performance.*
- *Technological capability is significantly associated with marketing performance.*

In addition, a case study was carried out to demonstrate how critical success factors can bring positive and substantive influence on the organisational performance in real-world markets. Pacific Basin is a leading shipping company headquartered in Hong Kong with a reputation for high-quality services to global clients and a strong commitment to sustainability. The company specialises in dry bulk shipping and operates a fleet of over 200 modern vessels with a global presence and a focus on innovation and excellence. The company's governance and performance were examined by the four critical success factors identified by this study (Appendix A). Initiatives in innovation and digitalisation led to efficient use of data and continuous business model adjustments. Furthermore, sufficient training and skills development bolstered solid risk governance capability and agile leadership. The company also continuously endeavours to improve technological capability in both vessel performance and voyage management. With these capabilities, the company has generated strong financial and operational performance, and marketing performance as well. As such, the case study manifested that the four constructs of critical success factors have effectively assisted Pacific Basin to achieve some valuable competitive advantages to improve organisational performance in the digital era.

## 6. Discussion

The findings of this study provide empirical, statistical and additional support to the six organisational theories that were found to be relevant. For example, the results of this study are aligned with one of the core tenets of Porter's generic strategies in that a cost leadership strategy is a critical success factor that a global shipping organisation needs in the digital era. This explains and clarifies its importance in a cyclical industry, as is the case of the global shipping industry with its brutal and extended downturns, and few and relatively rare market peaks [85]. The findings of this study also match with the resource-based view which helps explain the organisations which need resources that are aligned with the VRIO model (value (V), rareness (R), imitability (I) and organisational support (O)) in order that an organisation can use its internal resources, both tangible and intangible, in order to compete with its rivals for survival during the downturns and for maximising profits during periods of market booms.

This study also aligns with competence-based theory in that there is a structured and systemic method of strategic thinking on how an organisation can achieve high performance, surviving during the extended downturns and capitalising during the much shorter upcycles, over an extended period. The findings of this study also match with the dynamic capability theory that an organisation cannot only be seen to be optimising within the constraints of its strategy but also needs to have the ability to move the constraints through innovative strategic choices [75] while always pushing the boundaries on its financial capabilities, its cost leadership focus, its strategic capability and its customer service capability. In this study, having the strategic capability in preparing for the uncommon Knightian uncertainties which occur from time to time [86] in the global shipping industry in the digital era, such as business continuity planning, or in mitigating risks by preparing its teams to be market savvy in terms of positioning vessels ahead of market moves, can assist in reducing dependency on outside resources and/or resources that are not under an organisation's direct control.

In terms of the organisational culture theory, the relationship between culture and the effective functioning of an organisation has an important influence on its effectiveness [34]. The strategic capability in an organisation in the digital era includes areas of cost leadership

focus in the face of a plethora of digitalisation options that lack capability in tailoring to a global shipping organisation's specific need but are generalised towards a generic industrial base. How to capitalise on the opportunities that digitalisation promises without losing control of an organisation's cost focus is a key challenge which for an organisation culture which is frugal in its inception and in its values is a key challenge [87]. How to avoid the digitalisation traps that empower top managers and create an overreliance on quantitative data over holistic judgement and intuition [88] on the ground is a challenge all managers face in the digital era.

## 7. Conclusions

This study finds that there are four critical success factors of the global shipping industry in the digital era impacting three organisational performance measures that are relevant. This study is of practical value to global shipping practitioners in a number of ways. It provides empirical evidence to support the business facets that the creation, improvement and enhancement of the critical success factors of the global shipping industry can yield the outcomes of organisational performance that is desired with the deployment of the various tools and practices of global shipping management in the digital era. This study also emphasises reiterating the need for focus on formulating and executing the critical success factors of the global shipping industry in the digital era as relevant and as appropriately tailored to the specific shipping organisation so as to continuously manage and respond to the dynamic and ever-changing internal and external business environments in order to achieve improved organisational performance. Further, this study should assist global shipping practitioners in redesigning and reengineering their internal and external systems and processes by adjusting their primary areas of strategic focus and re-evaluating what is critical to their long-term organisational survival and profitable success in the global shipping industry in the digital era. Additionally, this study contributes to global shipping professionals in that the findings of the study provide some practical guidelines for the senior management to adopt, implement and execute the most appropriate global shipping strategy in order to identify and develop the critical success factors for the organisation. Senior managers in global shipping organisations can be guided by the findings in this study, and can develop, fine-tune, revise and enhance their own unique set of critical success factors that may be appropriate for their own circumstances.

This study addresses the complexities in the global shipping industry in the digital era by considering the critical success factors that are required to be successful in the industry. The insights thus found can broaden the understanding of the critical success factors of the global shipping industry and its organisational performance measures. However, this study is not immune from limitations. Due to the dynamic nature of the cyclical global shipping industry the critical success factors and the corresponding organisational performance measures of the global shipping industry in the digital era may vary over time and over cycles. Considering the total number of active shipping companies in the world, the sample size of this study is considered relatively small in size. This study obtained 109 valid survey responses and, although there is no definitive guide to determine and justify the sample size in an exploratory type of research, it appears that a larger sample size could very well yield more convincing results. The structured survey questionnaire adopted by this study and the results obtained are regarded as highly effective; however, both the findings and generalisability of this study might further be improved by involving a larger sample size. Another limitation of this study concerns its sample frame. The sample frame of this study was based and drawn on the membership of BIMCO, which means that the results of this study reflect the situation and thoughts of the BIMCO membership alone. In addition, it is possible that this study will somehow generate results that could well be different if different segments of the global shipping industry and different sub-sectors of the shipping industry were involved in this study. Other shipping groupings were not considered by this study as they were too specific or too much in a smaller non-representative segment of the global shipping industry and were thus excluded. The inclusion of these groups may well

have impacted on the final results. The experience and perception of the global shipping industry in the digital era may not be the same as some of the respondents who are yet to start on their digitalisation journey. Some global shipping organisations are investing vast sums of money on their digitalisation initiatives while at the other end of the scale others are still yet to begin their digitalisation journeys. This would have affected their responses and thus have affected the findings.

Another limitation of this study is that over 51% of the respondents in this study have been with their respective organisations for over 20 years. This could affect their view on digitalisation as 'just another fad' in shipping rather than a dramatic change in the way the entire business of global shipping is heading. A further limitation of this study is that, although the focus of this study is on global shipping, over 52% of the responses came from the Asia Pacific region while only 21% came from Europe and under 5% came from North America. There might be variations in formulating and deploying the approach in the management of the global shipping industry in the digital era which was not adequately captured by this study. The critical success factors of the global shipping industry in the digital era are in fact dynamic in nature in line with the fast pace of digitalisation in the industry and may well be better examined and reviewed in a longitudinal rather than a cross-sectional study. This study also did not consider possible mediators and moderators as it is exploratory in nature and the first of its kind in this field. Different constructs of the critical success factors of the global shipping industry in the digital era may be interrelated, and hence the moderation and mediation effect, if any, should be considered. In view of the complexity of the global shipping industry in the digital era, there may be antecedent variables, extraneous variables, confounding and intervening variables, for example cost leadership focus and financial capability may be antecedent variables. It would be very interesting for a new study to examine whether some of these relationships can be observed since this would certainly affect decision-making with regard to resource allocation. Other potential mediating variables include the size of the organisation [89] and the prevailing market conditions [38]. The inclusion of such mediating variables in future research is likely to provide further insights into the extent of the effect of these variables on the relationship between the critical success factors of the global shipping industry in the digital era and the relevant organisational performance measures. Therefore, future research can be carried out to include any possible moderator and/or mediator to examine the relationships between the critical success factors of the global shipping industry in the digital era and its organisational performance measures. Future researchers should consider investigating the causal relationships of different critical success factors of the global shipping industry in the digital era and the relevant organisational performance measures.

**Author Contributions:** Conceptualization, validation, and formal analysis, S.B., E.L. and T.L.Y. All authors have read and agreed to the published version of the manuscript.

**Funding:** This research was partially supported by a grant from the Hong Kong Polytechnic University (Project ID: P0039898 or project No: G-UAN9).

**Institutional Review Board Statement:** Not applicable.

**Informed Consent Statement:** Informed consent was obtained from all subjects involved in this study.

**Data Availability Statement:** Please refer to suggested Data Availability Statements in section "MDPI Research Data Policies" at https://www.mdpi.com/ethics (accessed on 23 March 2023).

**Conflicts of Interest:** The authors declare no conflict of interest.

## Appendix A

**Table A1.** Results of Hypothesis Testing.

| Model | Dependent Variable | Independent Variable | Value |
|:---:|:---:|:---:|:---:|
| 1 | Financial Performance (FIP) | Innovation Capability (INC) | b = 0.634 *** |
| | | Risk Governance Capability (RGC) | b = 0.260 *** |
| | | Leadership and Strategic Capability (LSC) | b = 0.415 *** |
| 2 | Operational Performance (OPP) | Risk Governance Capability (RGC) | b = 0.294 *** |
| | | Technological Capability (TEC) | b = 0.487 *** |
| 3 | Marketing Performance (MKP) | Technological Capability (TEC) | b = 0.307 *** |

*** $p \leq 0.001$.

**Table A2.** Results of Multiple Regression for Model 1: Financial Performance (FIP) as Dependent Variable for Hypothesis 1.

| Model Summary | | | |
|:---:|:---:|:---:|:---:|
| Model | | | |
| 1 | | R | 0.802 [a] |
| | | R squared | 0.643 |
| | | Adjusted R Squared | 0.63 |
| | | Standard Error of Estimate | 0.60847 |
| | | F | 46.928 |
| | | Sig. F | 0.000 [b] |

| Coefficients | | | | | |
|:---:|:---:|:---:|:---:|:---:|:---:|
| | Model | Unstandardised Coefficients | Standardised Coefficients | Collinearity Tolerance | VIF |
| 1 | (Constant) | 0.000 | | | |
| | INC | 0.634 *** | 0.634 | 1.000 | 1.000 |
| | RGC | 0.26 *** | 0.26 | 1.000 | 1.000 |
| | LSC | 0.415 *** | 0.415 | 1.000 | 1.000 |
| | TEC | −0.043 | −0.043 | 1.000 | 1.000 |

[a] Predictors: (constant), INC, RGC, LSC and TEC. [b] Dependent variable: FIP. *** $p \leq 0.001$.

**Table A3.** Results of Multiple Regression for Model 2: Operational Performance (OPP) as Dependent Variable for Hypothesis 2.

| Model Summary | | | | |
|---|---|---|---|---|
| Model | | | | |
| 2 | | R | | 0.591 [a] |
| | | R squared | | 0.349 |
| | | Adjusted R Squared | | 0.324 |
| | | Standard Error of Estimate | | 0.82212 |
| | | F | | 13.947 |
| | | Sig. F | | 0.000 [b] |
| **Coefficients** | | | | |
| Model | Unstandardised Coefficients | Standardised Coefficients | Collinearity Tolerance | VIF |
| 2 | (Constant) | 0.000 | | | |
| | INC | −0.047 | −0.047 | 1.000 | 1.000 |
| | RGC | 0.294 *** | 0.294 | 1.000 | 1.000 |
| | LSC | 0.153 | 0.153 | 1.000 | 1.000 |
| | TEC | 0.487 *** | 0.487 | 1.000 | 1.000 |

[a] Predictors: (constant), INC, RGC, LSC and TEC. [b] Dependent variable: OPP. *** $p \leq 0.001$.

**Table A4.** Results of Multiple Regression for Model 3: Marketing Performance (MKP) as Dependent Variable for Hypothesis 3.

| Model Summary | | | | |
|---|---|---|---|---|
| Model | | | | |
| 3 | | R | | 0.362 [a] |
| | | R squared | | 0.131 |
| | | Adjusted R Squared | | 0.097 |
| | | Standard Error of Estimate | | 0.95011 |
| | | F | | 3.91 |
| | | Sig. F | | 0.005 [b] |
| **Coefficients** | | | | |
| Model | Unstandardised Coefficients | Standardised Coefficients | Collinearity Tolerance | VIF |
| 3 | (Constant) | 0.000 | | | |
| | INC | 0.044 | 0.044 | 1.000 | 1.000 |
| | RGC | 0.144 | 0.144 | 1.000 | 1.000 |
| | LSC | 0.118 | 0.118 | 1.000 | 1.000 |
| | TEC | 0.307 *** | 0.307 | 1.000 | 1.000 |

[a] Predictors: (constant), INC, RGC, LSC and TEC. [b] Dependent variable: MKP. *** $p \leq 0.001$.

**Table A5.** Summary of the Critical Success Factors and Their Influence on the Organisational Performance of Pacific Basin.

| Critical Success Factors | Pacific Basin | Organisational Performance Measures |
| --- | --- | --- |
| Innovation Capability | Improved access to data and analytics drives its digitalisation initiatives, efficiency and optimisation initiatives driven by automation and improved process, new insights gained by better use of the combined big and small data, continuous business model adjustments with the insights gained | Financial Performance |
| Risk Governance Capability | Solid and well experienced skillset at board level, low Opex, low G&A costs, good audit functions to ensure controls are maintained well | Financial Performance and Operational Performance |
| Leadership and Strategic Capability | Diversity in board and senior management, staff internal transfer to develop management skills, good level of education and training budget, market savvy commercial teams | Financial Performance |
| Technological Capability | Continuous drive to get efficient ships, improve engine performance, improve voyage management by adjusting speed and trim during weather and different loading conditions | Operational Performance and Marketing Performance |

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
