# Peer review of "An Exploratory Study of the Critical Success Factors of the Global Shipping Industry in the Digital Era"

_jtaer, doi:10.3390/jtaer18020041_

Round 1

Reviewer 1 Report

The paper is generally well written and clear.  The topic is important dealing with the evolution of the global shipping industry in the digital era and the way shipping organisations will perform in the future. This work analysed four critical success factors of the global shipping industry in the digital era - innovation capability, risk governance capability, leadership and strategic capability and technological capability – and additionally resulted in three organisational performance measures that are useful for senior management teams - financial performance, operational performance and marketing performance. Overall, the exploration of the various research questions is interesting and the results are made clear to the reader. The literature review is comprehensive, but I believe that the paper should also make more extensive reference to the role of digitalisation in driving the maritime industry beyond its traditional ways of doing business and making its operations more efficient and sustainable. There is rich academic literature dealing with the impact of digitalization on the shipping industry that should be at least referred to the paper. Examples of previous relevant work are Ichimura, Y., Dalaklis, D., Kitada, M. and Christodoulou, A. (2022) Shipping in the era of digitalization: Mapping the future strategic plans of major maritime commercial actors. Digital Business 2(1), 100022. https://doi.org/10.1016/j.digbus.2022.100022; Aiello, G., Giallanza, A. and Mascarella, G. (2020) Towards shipping 4.0. A preliminary gap analysis. Procedia Manufacturing, 42 (2020), pp. 24-29. https://doi.org/10.1016/j.promfg.2020.02.019; De Andres Gonzalez, O., Koivisto, H., Mustonen, J.M and Keinänen-Toivola, M.M. (2021) Digitalization in just-in-time approach as a sustainable solution for maritime logistics in the Baltic Sea region. Sustainability, 13(3). https://doi.org/10.3390/su13031173; De la Peña Zarzuelo, I., Soeane, M.J.F. and Bermúdez, B.L. (2020) Industry 4.0 in the port and maritime industry: A literature review. Journal of Industrial Information Integration, 100173 (2020). https://doi.org/10.1016/j.jii.2020.100173; Dalaklis, D., Christodoulou, A., Ölcer, A.I., Ballini, F., Dalaklis, A. and Lagdami, K. (2021) The port of Gothenburg under the influence of the fourth stage of the industrial revolution: Implementing a wide portfolio of digital tools to optimize the conduct of operations. Maritime Technology and Research, 4(3), 253844. https://doi.org/10.33175/mtr.2022.253844. The implications of the paper’s results for further research are valuable as they highlight the areas that need to be further explored and open the way for further research in the field. I hope these comments are helpful to the authors and would assist them to improve their paper to make it suitable for publication in this journal.

Author Response

Thank you for your kind review. We have updated our article to reflect the key points of your review where appropriate and added some of the articles as suggested.

We appreciate your encouraging comments and find them very helpful.

Reviewer 2 Report

The topic of the research presented in the paper is relevant and useful for the Maritime Shipping industry, but the paper itself needs a deep review and reshape, especially in the Discussion section.

Discussion should present the principles, relationships and generalization coming from an analysis of the results. The current Discussion section (which is mixed with the Conclusions and should be separated) is a new list of findings of others authors and the relations of the results with those. It should be rewritten in order to focus on the authors findings and dry them down in terms of how they match with the hypothesis and how they answer the questions of the research. I don't mean that the authors should not interpret their findings in the light of previous studies, but they need to work harder on what they see from the results.

Apart of this, I find some missing studies when talking of the literature review: https://doi.org/10.1016/S0739-8859(07)21012-1 and https://doi.org/10.3390/app12052532. The first one is old but it is relevant to put in context what has been analyzed in the early days if digitalization. The second is new and have a collection of KPIs that can potentially relate with the Critical Success Factors from the present research.

I miss a definition of what the authors understand by Critical Success Factors.

When talking of the survey, how have the authors validated their questions previous to sending the survey? A common method is described in "Lawshe 1975. A quantitative approach to content validity. Personnel Psychology, 28: 563-575."

The authors mention they have reviewed a large number of consultancy reports. They should say which ones or at least they main ones.

The authors mention they've used for indicators Cronbach Alpha, one of them with 0.46 which, as they say, is very low. Please add information about the rest.

The authors mention at the end of the Results section they've used Case Study as part of their research without any information on the method use for it and no further information but the small piece included in Appendix A. Either they develop the case study here or they remove it and leave it for another paper.

Author Response

Thank you for your kind review. We have reviewed and updated our paper by separating the Discussions from the Conclusions, added some studies as recommended and improved our explanation of the results. We have also included a definition of Critical Success Factors. The validation was carried out by an expert panel, a pilot test and finally the final questionnaire. Consultancy reports were rather numerous and the limitations of the article would preclude itemising too many- however the key ones were included in the references and included in the literatire review. Case study referenece has been added.

Thank you for your efforts in improving our paper which is appreciated.

Reviewer 3 Report

Manuscript: An Exploratory Study of the Critical Success Factors of the Global Shipping Industry in the Digital Era

Structure and logic of manuscript; structure and research problems elaboration in the paper are correct and clear.

Abstract; well-structured however aim of the research should be clearly stated in the beginning of abstract. Also suggest skipping last keyword: digitisation

Introduction; must be supplemented by the aims of the study.

Literature overview and theoretical background; please consider changing the sequence in section name as follows; Theoretical background and literature overview because theoretical background of the study is valuable and rigorously elaborated. Also, research gap and research value added should be extended and specified thereof.

Research Framework and Hypothesis development; clear and logic research conceptualization followed by hypothesis settings with initially derived assumptions.

Method of research; consistent with clear assumptions and applied techniques; please re-edit the whole formula (409-414).

Results: in-depth analysis with revision of pre-defined hypothesis however results should also be referenced to the concrete tables in the Appendices.

Discussion and Conclusions: multifaced and with in-depth reference to the literature also with clear-stated limitations of study.

Summing-up I have some minor concerns (as stated above) to this valuable manuscript and recommend this paper for publishing however the entire manuscript must be edited anew according to requirements of MDPI publishing (please use word template of MDPI).

Author Response

Thank you for your kind review. 

We have considered your comments carefully and made the necessary changes as requested. We felt digitisation as an important keyword as it is often mistakenly taken as having the same meaning as digitalisation in order to access the key readership of the article to include shipping professionals (i.e. non-academics). We have updated the document in accordance with MDPI requirements and made further changes per your suggestions as required.

Thank you for your encouragement and kind review.

Reviewer 4 Report

Dear Authors,

The introduction section needs attention as the background of the study is not mentioned properly as well as this section lacks the proper sequence of information.  Some methodological insights should be added to the Introduction. Please review the literature gap analysis, assuring an internal coherence between research topic,
emerging findings, knowledge gaps and research goals (and then, research questions)

The literature review section needs to be expanded. Put your hypotheses into theory more strongly. Refer to the following research:

Jum, L., Zimon, D., Ikram, M., & Madzík, P. (2022). Towards a sustainability paradigm; the nexus between lean green practices, sustainability-oriented innovation and Triple Bottom Line. International Journal of Production Economics, 245, 108393.

Jum’a, L., Ikram, M., Alkalha, Z., & Alaraj, M. (2022). Do Companies Adopt Big Data as Determinants of Sustainability: Evidence from Manufacturing Companies in Jordan. Global Journal of Flexible Systems Management, 1-16.

Gajewska, T., Zimon, D., Kaczor, G., & Madzík, P. (2019). The impact of the level of customer satisfaction on the quality of e-commerce services. International Journal of Productivity and Performance Management.

etc.

The chain of evidence is weak in this research. The findings also must be better discussed in relation to the existing literature (targeted research gaps).

The analysis of results should be more deep.

Consclusion must be separated from Discussion – this is a common and strict requirement in all international journals. Conclusion must only describe and slightly generalize your findings. Discussion must 1) tell what do these findings mean and 2) put these findings and interpretations in the context of the international research

Good Luck!

Author Response

Thank you for your kind review.

In accordance with your review we have improved the introduction, literature review and included relevant research which was kindly pointed out by you. We have improved the chain of evidence links in order to better link it to the research gaps and separated the conclusions from the discussions. 

Thank you for your encouraging review which is appreciated.

Round 2

Reviewer 4 Report

The authors in practice did not refer to my earlier comments. Please re-read the first review and make the suggested changes. I will be able to make further comments after making the corrections.

Author Response

Thank you for your review and apologies for not better clarifying the changes that have been made. With the reviewers’ comments and recommendations, we have made the following changes:

  1. We have improved the background of the study by including the 3 key research questions to improve the clarity and the focus of the study. This provides clarity also on the sequencing while methodological insights were explained in the general research framework and linking it to the 3 research questions together with the explanation on the framework design choice.
  2. Literature gap analysis was improved by including additional research relevant to the study that has been suggested by the overall review panel which specifically relate to the study, namely; Critical Success Factors; Global Shipping Industry; and Digitalisation (or Digital Era). This has allowed us to improve the chain of evidence which we hope meets your recommendation of improving the paper.
  3. We have tightened the chain of evidence to make it better connected to the research methodology and improved the discussions of the findings with relation to the existing literature, especially the targeted research gaps).
  4. We have further deepened the analysis of the results and separated Conclusions from Discussions while improving the section on Discussions by telling what these findings mean and putting these findings and interpretations in the context of international research. We have considered shortening the Conclusions section without losing the important sub-section on limitations and felt that shortening it would not assist the study to be as robust as it could be and hence decided to retain it in its current form.

For the sake of good order we have highlighted the larger changes in line with your review. We hope that you find the changes appropriate and look forward to your further comments for improvement.

Round 3

Reviewer 4 Report

I have no more comments.

Author Response

1. The topic of the research presented in the paper is relevant and useful for the Maritime Shipping industry, but the paper itself needs a deep review and reshape, especially in the Discussion section. Discussion should present the principles, relationships and generalization coming from an analysis of the results. The current Discussion section (which is mixed with the Conclusions and should be separated) is a new list of findings of others authors and the relations of the results with those. It should be rewritten in order to focus on the authors findings and dry them down in terms of how they match with the hypothesis and how they answer the questions of the research. I don't mean that the authors should not interpret their findings in the light of previous studies, but they need to work harder on what they see from the results.
***
We have improved the background of the study by including the 3 key research questions to improve the clarity and the focus of the study. This provides clarity also on the sequencing while methodological insights were explained in the general research framework and linking it to the 3 research questions together with the explanation on the framework design choice.
***
2. Apart of this, I find some missing studies when talking of the literature review: https://doi.org/10.1016/S0739-8859(07)21012-1 and https://doi.org/10.3390/app12052532. The first one is old but it is relevant to put in context what has been analysed in the early days if digitalization. The second is new and have a collection of KPIs that can potentially relate with the Critical Success Factors from the present research. I miss a definition of what the authors understand by Critical Success Factors.
***
We have further deepened the analysis of the results and separated Conclusions from Discussions while improving the section on Discussions by better explaining what these findings mean and putting these findings and interpretations in the context of international research. We have considered shortening the Conclusions section without losing the important sub-section on limitations and felt that shortening it would not assist the study to be as robust as it could be and hence decided to retain it in its current form. Literature gap analysis was improved by including additional research relevant to the study that has been suggested by the overall review panel which specifically relate to the study, namely; Critical Success Factors; Global Shipping Industry; and Digitalisation (or Digital Era). This has allowed us to improve the chain of evidence which we hope meets your recommendation of improving the paper.
***
3. When talking of the survey, how have the authors validated their questions previous to sending the survey? A common method is described in "Lawshe 1975. A quantitative approach to content validity. Personnel Psychology, 28: 563-575." The authors mention they have reviewed a large number of consultancy reports. They should say which ones or at least they main ones. The authors mention at the end of the Results section they've used Case Study as part of their research without any information on the method use for it and no further information but the small piece included in Appendix A. Either they develop the case study here or they remove it and leave it for another paper.
***
The validation was carried out by an expert panel, a pilot test and finally the final questionnaire. Consultancy reports were rather numerous and the limitations of the article would preclude itemising too many- however the key ones were included in the references and included in the literature review. Case study reference has been added and the paragraph explanation of the study improved. We have added information to better explain the results and improve the link with the next section on Discussions and improved the wording to improve clarity.
***